Social calls in humpback whale mother-calf groups off Sainte Marie breeding ground (Madagascar, Indian Ocean)

Saloma Anjara 1 2 3
http://orcid.org/0000-0001-8916-237X Ratsimbazafindranahaka Maevatiana N. 1 2 3 maevatiana.ratsimbazafindranahaka@universite-paris-saclay.fr
Martin Mathilde 1
Andrianarimisa Aristide 2
Huetz Chloé 1
Adam Olivier 1 4
http://orcid.org/0000-0003-4873-2342 Charrier Isabelle 1
1 Institut des Neurosciences Paris-Saclay, Université Paris-Saclay, CNRS , Saclay , France
2 Département de Zoologie et Biodiversité Animale, Université d’Antananarivo , Antananarivo , Madagascar
3 Association Cétamada , Barachois Sainte Marie , Madagascar
4 Institut Jean Le Rond d’Alembert, Sorbonne Université , Paris , France
McElligott Alan
Electronic publication date: 2022 Aug 16
Publication date: 2022
Volume: 10
Electronic Location ID: e13785
Received 2022 Mar 9; Accepted 2022 Jul 5
Copyright: © 2022 Saloma et al.
Copyright year: 2022
Copyright holder: Saloma et al.
License: This is an open access article distributed under the terms of the Creative Commons Attribution License, which permits unrestricted use, distribution, reproduction and adaptation in any medium and for any purpose provided that it is properly attributed. For attribution, the original author(s), title, publication source (PeerJ) and either DOI or URL of the article must be cited.
License URL: https://creativecommons.org/licenses/by/4.0/

Keywords: Acoustic communication, Baleen whale, Vocal repertoire, Mother-offspring relationship

Funding: CNRS Total Foundation WWF Education Fund for Nature Program CeSigma Company This work was supported by the CNRS, the Total Foundation, the WWF Education Fund for nature program, and the CeSigma Company. The funders had no role in study design, data collection and analysis, decision to publish, or preparation of the manuscript.

==============================
Humpback whales (Megaptera novaeangliae) use vocalizations during diverse social interactions or activities such as foraging or mating. Unlike songs produced only by males, social calls are produced by all types of individuals (adult males and females, juveniles and calves). Several studies have described social calls in the humpback whale’s breeding and the feeding grounds and from different geographic areas. We aimed to investigate for the first time the vocal repertoire of humpback whale mother-calf groups during the breeding season off Sainte Marie island, Madagascar, South Western Indian Ocean using data collected in 2013, 2014, 2016, and 2017. We recorded social calls using Acousonde tags deployed on the mother or the calf in mother-calf groups. A total of 21 deployments were analyzed. We visually and aurally identified 30 social call types and classified them into five categories: low, medium, high-frequency sounds, amplitude-modulated sounds, and pulsed sounds. The aural-visual classifications have been validated using random forest (RF) analyses. Low-frequency sounds constituted 46% of all social calls, mid-frequency 35%, and high frequency 10%. Amplitude-modulated sounds constituted 8% of all vocalizations, and pulsed sounds constituted 1%. While some social call types seemed specific to our study area, others presented similarities with social calls described in other geographic areas, on breeding and foraging grounds, and during migrating routes. Among the call types described in this study, nine call types were also found in humpback whale songs recorded in the same region. The 30 call types highlight the diversity of the social calls recorded in mother-calf groups and thus the importance of acoustic interactions in the relationships between the mother and her calf and between the mother-calf pair and escorts.

Introduction

Baleen whales, like other cetaceans (toothed whales), use acoustic communication in many social contexts such as predator alert, foraging cooperation, mating, individual identification, and parental care (Edds-Walton, 1997; Tyack, 1999; Dudzinski, Thomas & Gregg, 2009). Sounds used by baleen whales include vocalizations and sounds generated by active surface behaviors (breaching, tail or pectoral fins slapping, etc.). Some baleen whales’ vocalizations, attributed to males, are organized in a repeated manner and in long bouts to constitute what is called a ‘song’ (Edds-Walton, 1997; Tyack, 1999; Dudzinski, Thomas & Gregg, 2009). Baleen whales also produce non-song vocalizations called ‘social calls’ (Payne, 1978; Tyack, 1981; Silber, 1986). Unlike songs, social calls have been described as variable through time, interrupted by silent periods, apparently unpredictable, and not showing the rhythmic, consistent and continuous temporal pattern of songs (Tyack, 1981; Silber, 1986). While all social groups in all species of baleen whales potentially produces social calls, social calls produced by mother-calf pairs have been proposed to have the particularity of being produced at a low rate and/or at a low amplitude to minimize the risk of alerting predators (Videsen et al., 2017; Nielsen et al., 2019; Parks et al., 2019).

Humpback whales are baleen whales known for their complex, highly structured, and organized songs produced by males in breeding areas (Payne & McVay, 1971) but they also generate social calls. Humpback whales’ social calls have been the subject of several studies, both in feeding or breeding grounds, as well as on migration routes (D’Vincent, Nilson & Hanna, 1985; Silber, 1986; Cerchio & Dahlheim, 2001; Dunlop et al., 2007; Dunlop, Cato & Noad, 2008; Zoidis et al., 2008; Stimpert, 2010; Stimpert et al., 2011; Rekdahl et al., 2013; Fournet, Szabo & Mellinger, 2015; Rekdahl et al., 2017; Epp et al., 2021; Epp, Fournet & Davoren, 2021; Indeck et al., 2021). Distinct acoustic units of vocalization or “call types” were generally defined either using an automatic clustering method based of several acoustic parameters (Stimpert, 2010; Stimpert et al., 2011; Fournet, Szabo & Mellinger, 2015) or an aural-visual classification (Dunlop et al., 2007; Dunlop, Cato & Noad, 2008; Zoidis et al., 2008; Rekdahl et al., 2013, 2017; Fournet, Szabo & Mellinger, 2015; Fournet et al., 2018; Epp et al., 2021; Epp, Fournet & Davoren, 2021; Indeck et al., 2021) validated using a supervised automatic classification (e.g., discriminant function analysis–DFA, Classification and Regression Tree–CART, or random forest–RF) or not. The recordings were obtained using moored or towed hydrophones (Dunlop et al., 2007; Dunlop, Cato & Noad, 2008; Stimpert, 2010; Rekdahl et al., 2013, 2017; Fournet, Szabo & Mellinger, 2015; Fournet et al., 2018; Epp et al., 2021; Epp, Fournet & Davoren, 2021), using hydrophone carried by divers (Zoidis et al., 2008), or using animal borne tags (Stimpert, 2010; Stimpert et al., 2011; Indeck et al., 2021).

Humpback whales’ social calls vary from low to high-frequency calls and differ in their general structure (Silber, 1986; Dunlop et al., 2007; Rekdahl et al., 2013). Social calls can be categorized into a broad class or category according to their general characteristics (e.g., Dunlop et al., 2007; Zoidis et al., 2008; Fournet et al., 2018; Epp, Fournet & Davoren, 2021). Humpback whale vocal repertoire has been described as a mix of discrete call types (call types acoustically distinct from one another and with little variation) and graded call types (call types with varying characteristics that form a continuum and represented by some common cases and by intermediate forms) (Epp, Fournet & Davoren, 2021). The variation of call types in humpback whale is likely related to factors such age, sex, or body mass (Cerchio & Dahlheim, 2001; Epp, Fournet & Davoren, 2021).

Social calls are produced by the same sound generator as for songs, located inside the respiratory system, composed of the lungs (air source), the laryngeal sac (air source and acoustic resonator), and the nasal cavities (another resonator) (Adam et al., 2013). Individuals in all group compositions produce social calls: lone individuals, multiple animals, pairs, and mother-calf pairs accompanied or not by escorts (Dunlop, Cato & Noad, 2008).

Social calls in adult humpback whales were first described as ranging from 50 Hz to over 10 kHz with fundamental frequencies below 3 kHz (Silber, 1986). These social calls were reported to be produced when whales are predominantly in groups of three or more adults, in surface-active groups including both females and males (Silber, 1986). Although up to now only few direct quantitative comparisons were conducted and most comparisons were based on aural and visual characteristics, several studies suggested that some calls types are shared between humpback whales from different regions (Dunlop et al., 2007; Stimpert et al., 2011; Fournet, Szabo & Mellinger, 2015; Rekdahl et al., 2017; Epp, Fournet & Davoren, 2021).

The first confirmed recording of humpback whale calves’ social calls was performed by Zoidis et al. (2008) using a technique involving a two-element hydrophone array. It has been suggested that calves’ vocal repertoire is limited and vocalizations are simple in structure, short, predominantly composed of low frequencies, and have a relatively narrow frequency bandwidth (Zoidis et al., 2008). Humpback whales are known to be vocal learners (Tyack, 1998; Janik, 2014). Although mostly described for male songs, the vocal learning likely also apply to social calls. Calf might have to learn and master their calls, or aspects of them (Tyack, 1998; Epp, Fournet & Davoren, 2021).

Up to now, the biological functions of social calls remain unclear. Visual observations from surface activity suggested that social calls could serve either as a sign of aggression among males competing for the ‘principal escort’ status (Tyack, 1983; Baker & Herman, 1984; Silber, 1986) or, in some specific breeding competitive contexts, depending on sex and group composition, among adults as a deterrent to dissuade approach from other whales of same or opposite sex (Tyack, 1983).

Mother-calf pairs constitute the only stable social association on breeding grounds in humpback whales. Calves are born in a rich acoustic environment filled with songs and social calls. Recent acoustic studies on the social calls of humpback whales investigated vocal production from mother-calf pairs (Dunlop et al., 2007; Fournet, Szabo & Mellinger, 2015; Recalde-Salas et al., 2020; Indeck et al., 2021) with an attempt to assign social calls to females or calves. Identifying the sound source remains a real challenge (Saddler et al., 2017; Indeck et al., 2021), and thus further investigations are needed to accurately assign social call types to mothers or calves.

The social calls of humpback whales in the South Western Indian Ocean remained poorly documented. This present study aimed to investigate the repertoire of social calls recorded by suction-cup acoustic tags (Acousonde 3B) attached to female humpback whales or their calf off Sainte Marie island, Madagascar, South Western Indian Ocean, during four breeding seasons. Our goal was to complement the knowledge of acoustic communication in poorly documented yet critical social groups such as mother-calf groups and to contribute to the global catalogue of humpback whale social calls. The social calls described in breeding and foraging grounds are likely to be different, as they are produced in different behavioral contexts. Such knowledge is thus crucial to better understand the biological function of social calls, but also to investigate dialects and thus vocal learning of social calls. In addition, the description of the repertoire of social calls in mother-calf groups is a crucial material for investigating the role of acoustic communication in humpback whale mother-calf interactions in the future.

Materials and Methods

Study area

We collected acoustic data during winters 2013, 2014, 2016, and 2017 in the coastal waters off Sainte Marie island, Madagascar, South Western Indian Ocean (between latitudes 17° 19′ and 16° 42′ South, and longitudes 49° 48′ and 50° 01′ East), where mother-calf pairs come in these relatively calm and shallow waters (Trudelle et al., 2016).

Data collection and tagging procedure

We used Acousonde 3B (www.acousonde.com) attached to females or calves via four suction cups, a non-invasive attachment system, to record sounds. We deployed tags from a 6.40 m rigid motorboat using a 5 m handheld carbon-fiber pole. The boarding team consisted of staff experienced in successfully approaching mother-calf pairs: one operator (boat pilot), photographer, note-taker, and tagger. While the boarding team may change between outings, the team composition was always respected and each team member held one role within a whole outing. The tags were placed on the back, near the dorsal fin of the animal. Calves were tagged using one of the two approaches described in Stimpert et al. (2012) and in Huetz et al. (2022) in order to minimize disturbance to the mother-calf pair. Tagging efforts were terminated if the pair displayed avoidance behavior or if the calf was not successfully tagged within 30 min. Immediate behavioral response of the animals to tagging was recorded as in Stimpert et al. (2012). All mother-calf pairs were photo-identified to avoid double-sampling within the calving season. Tagged animals were not followed after tag deployment to avoid any further disturbance of their behavior. After tag deployment or an aborted attempt, the boat slowly moved away in the opposite direction of the mother-calf pair. The tag was retrieved after few hours or the following day, when it detached itself from the animal (usually as a consequence of rubbing against the mother, surface active behavior, etc.). The Fisheries Resources Ministry, Madagascar, fully approved all experimental protocols (Research and Collect permits #44/13-MPRH/SG/DGPRH, #43/14-MRHP/SG/DGRHP, #28/16-MRHP/SG/DGRHP, and #26/17-MRHP/SG/DGRHP). This present study complies with the European Union Directive on the protection of animals used for scientific purposes (EU Directive 2010/63/EU).

For each spotted group with mother-calf pair, approaches similar to touristic boats, complying with Madagascar’s Code of Conduct for whale watching activities (Inter-ministerial decree March 8th, 2000), were adopted: the boat’s speed was reduced gradually at 800 m distance from the spotted group. The observation area for mother-calf groups was set at a 200 m radius around them. We used this distance to observe the groups before tag deployment. We noted the group composition: lone mother-calf pair (MC) or mother-calf pair accompanied by one or several escorts (MCE). Behavioral observations and photo-identification were obtained concurrently for each group, and the calf’s relative age was estimated using the angle of furling of its dorsal fin (Cartwright & Sullivan, 2009; Huetz et al., 2022): C1 (neonate)–calf presenting some folds, scars, and skin color that tends to be light grey dorsally and white ventrally and with less than 44° dorsal fin furl (Faria et al., 2013), C2: very young but non-neonate calves having more than 45° but less than 72° dorsal fin furl, and C3: older calves that have unfurled dorsal fins (>72°). Depending on the opportunity and on how the group behaved, we tagged the calf or the mother using either a passive or active approach, as described in Stimpert et al. (2012) and Huetz et al. (2022).

We did not follow the animals after tag deployment to avoid further disturbance of their behavior. This means that change in group composition during the data recording is unknown. MC could be joined by other escort(s) or escort(s) in MCE might have left. Our tagging processes lasted 21 min on average and never exceeded 30 min (i.e., our maximum duration to tag a whale). Beside the mother-calf recordings, we also punctually recorded several male songs using a towed Aquarian H2a hydrophone deployed from the boat (engine off).

Vocal repertoire and acoustic analysis

We downloaded the audio files from the tags as *.MT files, converted them to *.WAV format using GoldWave software (GoldWave Inc., St. John’s, Canada), and analyzed them using Avisoft SASLab Pro version 5.207 (Avisoft Bioacoustics, Nordbahn, Germany). We produced spectrograms of the acoustic recordings using 1,024-point Fast Fourier Transform, 75% overlap, and Hamming window.

Social calls that were clearly audible and distinguishable were aurally and visually identified, classified, and then compared with social call catalogues available in the literature (Dunlop et al., 2007; Dunlop, Cato & Noad, 2008; Zoidis et al., 2008; Stimpert et al., 2011; Rekdahl et al., 2013, 2017; Fournet, Szabo & Mellinger, 2015; Epp, Fournet & Davoren, 2021; Indeck et al., 2021). Call types qualitatively similar to call types described in these catalogues (reference catalogue hereafter) were given the same name. New names were assigned for the remaining call types onomatopoeically, as the behavioral context or biological function of these new call types is still unknown and the main acoustic structure can be common to several call types (e.g., several call types can be low-frequency upsweep calls). Distant song units could not be confused with social calls in our dataset as songs can be easily recognized by their well-organized temporal pattern. Additionally, we used 16 of the recorded male songs to determine if some social calls were similar to males’ song units.

For each identified social call, we measured nine temporal and spectral characteristics (Table 1). Similar to Dunlop et al. (2007), we categorized social calls as either low-frequency sounds (LF), mid-frequency harmonic sounds (MF), high-frequency harmonic sounds (HF), amplitude modulated sounds (AM), noisy and complex sound (NC), or pulsed sounds (PS). LF corresponds to calls with peak frequency below 160 Hz. MF corresponds to calls with a peak frequency ranging from 170 to 550 Hz. HF corresponds to calls with a peak frequency above 700 Hz. AM corresponds to sounds consisting of a combination of long harmonic and amplitude modulated components with peak frequency ranging from 20 to 300 Hz. NC corresponds to broadband calls or harmonic calls with additional noise-like features. PS corresponded to low-frequency sounds repeated rhythmically. Representative spectrograms (Hamming window, FFT window size: 1,024 pts, 90% overlap) for the identified social calls were generated using the Seewave package (Sueur, Aubin & Simonis, 2008) in R software (R Core Team, 2021).

Table 1 Measured temporal and spectral characteristics for each identified social call.

Measurements	Abbreviation	Description	
Manual measurements			
Duration (s)	Dur	Total duration of the social call (a)	
Fundamental frequency (Hz)	F0	Fundamental frequency (b)	
Frequency excursion (Hz)	Fexc	Difference between the higher and lower frequencies (c)
Measured only when applicable
For harmonic-structured calls, Fexc was measured on the first visible frequency band	
Automatic measurements			
Peak frequency (Hz)	Fmax	Frequency at which the maximum amplitude level occurs (b)	
1st energy quartile frequency (Hz)	Q25	Frequency below which 25% of the total energy occurs (b)	
2nd energy quartile frequency (Hz)	Q50	Frequency below which 50% of the total energy occurs (b)	
3rd energy quartile frequency (Hz)	Q75	Frequency below which 75% of the total energy occurs (b)	
Frequency bandwidth (Hz)	Bdw	Frequency bandwidth within which the total energy fell within 12 dB of Fmax (b)	
Pulse rate (Hz)	PR	Number of pulse per second as measured using the pulse train analysis function in Avisoft SASLab Pro
PR was considered to be zero for calls without a pulsed structure	
Note:

(a) Measured on the oscillogram. (b) Measured on the averaged spectrum. (c) Measured on the spectrogram.

To obtain an overview of the intensity of sounds recorded in mother-calf groups, we measured the received level (in dB re 1μPa RMS) of the most common and aurally and visually easily identifiable call types using the Root Mean Square (RMS) function in Avisoft SASLab Pro. We considered only 10 good quality calls with a signal-to-noise ratio above 10 dB and without overlap for each selected call type. Using such criteria, calls selected for such measurements are likely produced by individuals constituting the focal mother-calf group (i.e., the mother, the calf, and possibly the escort if present). Mother-calf pair don’t gather with other pairs, so the only calls detected from our recordings can only come from individuals of the focal groups. Songs are highly different, so no confusion could be made. A similar procedure was used previously on humpback whale mother-calf pairs in Western Australia (Videsen et al., 2017).

Statistical analysis

To validate our aural-visual classification of calls by categories and by types, we performed random forest (RF) analyses (Thiebault et al., 2019; Indeck et al., 2021) using the randomForest (Liaw & Wiener, 2002) package in R. We calculated the global accuracy of the RF classification of calls by categories and by types (defined as accuracy = 1 − OOB, where OOB stands for out-of-bag error, the misclassification error rate) and computed the Gini index, which gives the importance of the variables used for the classification. We only included acoustic variables that were primarily measurable for most of the calls: Dur, Fmax, Q25, Q50, Q75, and PR. We included only call types for which we had at least six exemplars. The number of variables to be randomly selected at each split was set at 2 (as we had only six variables), and the number of trees grown was set at 500. We used a balanced RF design to maintain equal sample sizes of each category or type in the classification and avoid the over-representation of the most represented classes (Chen, Liaw & Breiman, 2004). In this design, each tree of the RF is built with the same number of calls per category or per type (i.e., the smallest number of calls for a given category or type).

Results

Tag deployments

We performed 62 successful tag deployments (35 on calves and 27 on mothers) during the four years of data collection. Acoustic data were usable for 21 deployments (21 different mother-calf groups): seven deployments on mothers and 14 deployments on calves (Table 2). The other deployments were not analyzed due to the high background noise level or their short durations (less than 30 min). Background noise was mainly present when the tag was placed in a higher position close to the dorsal fin and thus often out of the water, especially on calves for which surfacing activities occurred very often. Of the 21 studied groups, we identified 12 as MC and nine as MCE. Three groups had a C1 calf, three had a C2 calf, and 15 had a C3 calf. We detected social calls in all of the studied groups.

Table 2 Details of the tagging sampling effort. MC: lone mother-calf pair. MCE: mother-calf pair accompanied by one or several escorts.

Date	Tagged individual	Group type	Calf’s relative age	Analyzed recording duration (hh:mm)	
07/08/2013	Mother	MCE	C3	04:05	
16/08/2013	Mother	MCE	C1	01:22	
05/09/2013	Mother	MC	C3	01:26	
09/09/2013	Mother	MCE	C3	00:39	
12/09/2013	Calf	MC	C3	03:14	
15/09/2013	Calf	MCE	C3	01:06	
05/08/2013	Mother	MC	C3	05:15	
24/08/2014	Mother	MC	C3	02:00	
26/08/2014	Mother	MCE	C2	14:23	
29/08/2014	Calf	MC	C3	00:27	
08/09/2014	Calf	MCE	C3	03:27	
09/09/2014	Calf	MC	C3	05:00	
10/09/2014	Calf	MC	C1	02:15	
11/09/2014	Calf	MCE	C2	00:35	
17/09/2014	Calf	MC	C3	00:28	
11/08/2016	Calf	MCE	C2	05:38	
17/08/2016	Calf	MCE	C3	07:14	
18/08/2016	Calf	MC	C1	10:14	
05/09/2016	Calf	MC	C3	07:46	
28/08/2017	Calf	MC	C3	05:32	
01/09/2017	Calf	MC	C3	02:30	
Note:

C1 (neonate): calf presenting some folds, scars, and skin color that tends to be light grey dorsally and white ventrally and with less than 44° dorsal fin furl. C2: very young but non-neonate calves having more than 45° but less than 72° dorsal fin furl. C3: older calves that have unfurled dorsal fins (>72°).

Social call classification

A total of 2,033 social calls were clearly distinguishable. Aural-visual characteristics allowed the classification of these calls into 30 call types representing five of the six main categories suggested by Dunlop et al. (2007) (Table S1): low-frequency sounds (LF), mid-frequency harmonic sounds (MF), high-frequency harmonic sounds (HF), amplitude modulated sounds (AM), and pulsed sounds (PS). We did not find any call corresponding to the noisy and complex sound (NC) category. Eleven call types were named after qualitatively similar calls in the reference catalogue we used and 17 were given new name as they appeared qualitatively different (Table S1). Audio examples of call types are provided (Audio S1).

Low-frequency sounds (LF)

LF was the most represented category (46%, N = 925/2,033). Eleven call types were within the LF category: “100 Hz sound”, “bass”, “boom”, “gru”, “snort”, “burp”, “guttural”, “thowp”, “wop”, “bark”, and “drum” (Fig. 1). “Bass” and “wop” (Figs. 1B and 1I respectively) were the most common LF calls (heard in eight groups each), followed by “100 Hz” and “thowp” (Figs. 1A and 1H) (seven groups), “gru” and “snort” (Figs. 1D and 1E) (six groups), and by “boom” (Fig. 1C) (four groups). The remaining LF calls were rare (heard in two groups for “drum”, Fig. 1K, and only one group for “guttural”, “burp”, and “bark”, Figs. 1G, 1F and 1J respectively). “Burp”, “bark”, and “drum” were only heard in MC groups. The remaining LF calls were heard in both MC and MCE groups. “Bass” (Fig. 1B) was a harmonic sound with a fundamental frequency below 40 Hz on average, and can sometimes be masked by background noise. “Wop” (Fig. 1I) and “thowp” (Fig. 1H) were brief harmonic upsweep sounds similar in frequency but different in duration. The “100 Hz” call (Fig. 1A) was a long, relatively flat call. “Gru” (Fig. 1D) was a short harmonic sound like “snort” but with more spaced harmonics. “Snort” (Fig. 1E) can be produced in sequences. “Boom” (Fig. 1C) was a harmonic sound produced either in sequence or alone. “Boom” was frequently produced in series with “100 Hz” following a well-defined order (100 Hz–boom–boom–100 Hz). “Guttural” (Fig. 1G) appeared to be a “composite call” consisting of non-overlapping “gru” and “heek” (MF) not separated with a silence. “Drum” (Fig. 1K) was a very short call always produced in series, and “burp” (Fig. 1F) was a short harmonic sound with several close harmonics. “Bark” (Fig. 1J) was a short harmonic sound with ascending frequency modulation.

Figure 1 Spectrograms of low-frequency sounds (LF).

(A) 100 Hz, (B) bass, (C) boom, (D) gru, (E) snort, (F) burp, (G) guttural sound, (H) thwop, (I) wop, (J) bark, (K) drum. Most of the LF sounds were harmonic-structure sounds, produced alone, except for boom and snort that can be produced in sequences. Drum sounds were always produced in series. Spectrogram parameters: Hamming window, FFT window size: 1,024 pts, 90% overlap. Generated using the Seewave package in R.

Mid-frequency harmonic sounds (MF)

MF was the second most represented category (35%, N = 718/2,033). Eight call types were within the MF category: “groan”, “downsweep”, “woohoo”, “trumpet”, “heek”, “whoop”, “wiper”, and “creak” (Fig. 2). “Heek” was the most common MF call (heard in nine groups), followed by “whoop” (seven groups), “downsweep” and “woohoo” (five groups), and by “trumpet” (three groups). “Groan”, “wiper”, and “creak” were uncommon (heard in one group only) and were only heard in MC groups. The remaining MF calls were heard in both MC and MCE groups. “Heek” (Fig. 2E) was a short MF call with a variable frequency modulation pattern (ascending, descending, or modulated). In some instances, heek was produced with “gru” (LF), with a short silence separating the two vocalizations to constitute a “combined call”. “Whoop” (Fig. 2F) was a long upsweep call starting with a flat part and fast ascending frequency. “Downsweep” (Fig. 2B) was a long call showing a descending frequency slope with well-spaced harmonics. “Woohoo” (Fig. 2C) was a long-duration call (i.e., several seconds) showing a variable frequency-modulated pattern. “Trumpet” (Fig. 2D) was a call produced alone or associated with “gru” or “slight snort”. “Downsweep”, “woohoo”, and “trumpet” were found in humpback whale songs recorded around the study site. “Groan” (Fig. 2A) was a long harmonic call. “Wiper” (Fig. 2G) was a short harmonic sound with a U-shape frequency modulation pattern, always produced in series (four to five repetitions) but with a random temporal pattern. “Creak” (Fig. 2H) was a composite call constituted by two different non-overlapping successive sounds without silence, and it showed the widest frequency bandwidth.

Figure 2 Spectrograms of mid-frequency harmonic sounds (MF).

(A) Groan, (B) downsweep, (C) woohoo, (D) trumpet, (E) heek, (F) whoop, (G) wiper, (H) creak. Downsweep, woohoo and trumpet calls were also found in humpback whale songs recorded around the study site. Heeks were produced in association with Grus (LF) in some instances. Spectrogram parameters: Hamming window, FFT window size: 1,024 pts, 90% overlap. Generated using the Seewave package in R.

High-frequency harmonic sounds (HF)

HF was the third most represented category (10%, N = 202/2,033). Two social calls were within this category: “squeak” and “ascending shriek” (Fig. 3). Both were quite common: squeak (Fig. 3A) was heard in seven groups, and “ascending shriek” (Fig. 3B) was heard in four groups. “Squeak” and “ascending shriek” were heard in both MC and MCE groups. “Squeak” was a very short call with frequencies above 1 kHz, and “ascending shriek” was one of the longest social calls with the highest frequencies amongst all. Both call types were also found in humpback whale songs recorded around the study site.

Figure 3 Spectrograms of high-frequency sounds (HF).

(A) Squeak, (B) ascending shriek. Squeaks were very short calls with frequencies above 1 kHz and ascending shrieks were among the longest social call types with the highest frequencies among all call types. Spectrogram parameters: Hamming window, FFT window size: 1,024 pts, 90% overlap. Generated using the Seewave package in R.

Amplitude modulated sounds (AM)

AM was the fourth most represented category (8%, N = 167/2,033). Five social calls were within the AM category: “door”, “whine”, “trill”, “bug sound”, and “AM grunt” (Fig. 4). “AM grunt” (Fig. 4E) and trill (Fig. 4C) were the most common MF calls (heard in six and five groups, respectively). The remaining calls were relatively uncommon since they were heard only in one group each. “Door” (Fig. 4A) was heard in an MC group. “Whine” (Fig. 4B) and bug (Fig. 4D) were found in MCE groups only. “Trill” and “AM grunt” were heard in both MC and MCE groups. “Trill”, “door”, “whine”, and “bug” are long calls ranging from 1 to 5 s duration with a peak frequency ranging from 100 to 400 Hz. They were produced in bouts of random durations. “AM grunt” was short and, while commonly produced alone, it was sometimes associated with LF calls such as “gru” or “snort”. “Whine”, “trill”, and “bug” were also found in humpback whale songs recorded around the study site.

Figure 4 Spectrograms of amplitude-modulated sounds (AM).

(A) Door, (B) whine, (C) trill, (D) bug sound, (E) AM grunt. Except for AM grunts, AM sounds are long calls ranging from 1 to 5 s with a peak frequency between 100 and 400 Hz, produced in bouts of random durations. Whine, trill, and bug sounds were also found in humpback whale songs recorded around the study site. Spectrogram parameters: Hamming window, FFT window size: 1,024 pts, 90% overlap. Generated using the Seewave package in R.

Pulsed sounds (PS)

PS was the least represented category (1%, N = 21/2,033). Four social calls were classified in this category: “fry”, “bubble sound”, “moped”, and “gloop” (Fig. 5). These calls consisted of a repetition of very short, low-frequency sounds, and they were pretty uncommon. “Fry” (Fig. 5A) was heard in two groups and the remaining PS calls were heard in only one group each. “Fry” was heard in both MC and MCE groups. “Bubble sound” (Fig. 5B) was heard in a MC group. “Moped” (Fig. 5C) and “gloop” (Fig. 5D) were only found in MCE groups.

Figure 5 Spectrograms of pulsed sounds (PS).

(A) Fry, (B) bubble sound, (C) moped, (D) gloop. PS are repetitive short and low frequency sounds. Spectrogram parameters: Hamming window, FFT window size: 1,024 pts, 90% overlap. Generated using the Seewave package in R.

Aural-visual classification validity

For the validation of our classification RFs, 757 social calls representing 17 call types were used for the analyses (social calls for which Dur, Fmax, Q25, Q50, Q75, and PR were all measured). These analyzed social calls exclude call types for which we had less than six exemplars. The analyzed call types were: 100 Hz, bass, boom, gru, snort, burp, thowp, wop, downsweep, woohoo, trumpet, heek, whoop, squeak, ascending shriek, trill, and fry. The RF showed a global accuracy of prediction of 93% for classifying the calls into the five defined main categories (OOB error rate = 7%, Table 3). The acoustic variables showing the highest importance for the classification were Q25, Fmax, and PR (Gini index: 10.24, 10.18, and 6.93 respectively, Table 3). Most call categories showed low individual classification error rates. For classifying the calls by types, the RF showed a global accuracy of prediction of 77% (OOB error rate = 23%, Table 4). The acoustic variables showing the highest importance for classification were Fmax, Q25, and Dur (Gini index: 32.10, 31.78, and 26.94 respectively, Table 4). Of the 17 calls included in the RF analysis, only four call types showed exceptionally high error rates (≥50%): gru, snort, thowp, and wop. These call types may share features with other call types (short, harmonic, and low-to-medium frequency calls).

Table 3 Random forest classification matrix for mother-calf groups’ social call categories.

		Predicted class		
		AM	HF	LF	MF	PS	Error	
True class	AM	21	0	0	2	0	0.09	
HF	0	94	0	0	0	0	
LF	0	0	438	16	0	0.04	
MF	17	7	8	143	0	0.18	
PS	0	0	0	0	11	0	
Variables	Q25	Fmax	PR	Dur	Q50	Q75	
Gini index	10.24	10.18	6.93	6.85	5.1	4.68	
Note:

The overall error rate (out-of-bag error rate, OOB) was 7%. The last column indicates the classification error for each main call category. The bottom lines show the used acoustic variables along with the Gini index reflecting their relative importance in the classification.

Table 4 Random forest classification matrix for mother-calf groups’ social call types.

		Predicted class	
		100 Hz	Bass	Boom	Gru	Snort	Burp	Thowp	Wop	Downsweep	Woohoo	Trumpet	Heek	Whoop	Squeak	Ascending shriek	Trill	Fry	Error	
True class	100 Hz	132	0	7	0	4	0	0	0	0	0	0	8	0	0	0	0	0	0.13	
Bass	0	49	0	4	1	2	0	1	0	0	0	0	0	0	0	0	0	0.14	
Boom	18	0	121	0	3	0	0	1	0	0	0	13	0	0	0	0	0	0.22	
Gru	0	1	1	6	0	1	3	0	0	0	0	0	0	0	0	0	0	0.5	
Snort	4	0	0	2	21	0	8	4	1	0	0	3	0	0	0	0	0	0.51	
Burp	0	0	0	1	0	12	0	1	0	0	0	0	0	0	0	0	0	0.14	
Thowp	1	0	1	1	5	0	0	1	0	0	0	0	0	0	0	0	0	1	
Wop	0	1	0	0	2	1	2	5	0	1	0	0	0	0	0	0	0	0.58	
Downsweep	0	0	0	0	0	0	0	0	27	8	0	0	5	0	1	1	0	0.36	
Woohoo	0	0	0	0	0	0	0	0	6	18	0	0	0	0	0	4	0	0.36	
Trumpet	0	0	0	0	0	0	0	0	0	0	10	0	0	0	0	3	0	0.23	
Heek	2	0	0	0	0	0	0	0	0	0	0	11	0	1	2	0	0	0.31	
Whoop	0	0	0	0	0	0	0	0	6	4	2	3	61	0	0	0	0	0.2	
Squeak	0	0	0	0	0	0	0	0	0	0	0	1	0	68	7	0	0	0.11	
Ascending shriek	0	0	0	0	0	0	0	0	0	0	0	0	0	2	16	0	0	0.11	
Trill	0	0	0	0	0	0	0	0	2	2	1	0	0	0	0	18	0	0.22	
Fry	0	0	0	0	0	0	0	0	0	0	0	0	0	0	0	0	11	0	
Variables	Fmax	Q25	Dur	Q50	Q75	PR	
Gini index	32.10	31.78	26.94	24.78	20.74	7.52	
Note:

The overall error rate (out-of-bag error rate, OOB) was 23%. The last column indicates the classification error for each main call type. The bottom lines show the used acoustic variables along with the Gini index reflecting their relative importance in the classification.

Received level

Five call types were selected for the calculation of the received level: three LF (“100 Hz”, “bass”, and “boom”), one AM (“trill”), and one MF sound (“heek”). The received level ranged from 132 to 154 dB re 1μPa RMS with an average of 141 dB re 1μPa RMS (N = 50, 10 per call type; Table 5).

Table 5 Received level measured for the most common aurally and visually easily identifiable call types.

Calls types	Mean ± SD amplitude received level
(in dB re 1μPa RMS)	N	Tagged individual	
100 Hz	141 ± 4	10	Mother	
Bass	154 ± 6	10	Mother	
Boom	135 ± 3	10	Mother	
Trill	132 ± 2	10	Mother	
Heek	145 ± 6	10	Calf	

Discussion

Humpback whales’ vocal activity is well known for its diversity and complexity at individual, group, and population levels. Social calls occur in all group compositions, in both breeding and foraging grounds as well as on migratory routes (Dunlop et al., 2007; Dunlop, Cato & Noad, 2008; Zoidis et al., 2008; Stimpert, 2010; Rekdahl et al., 2013; Fournet, Szabo & Mellinger, 2015; Recalde-Salas et al., 2020; Epp et al., 2021; Epp, Fournet & Davoren, 2021; Indeck et al., 2021). The aural-visual analysis allowed us to identify 30 social calls distributed into five main call categories (LF, MF, HF, AM, PS) for mother-calf groups off Sainte Marie island. Our RF analyses showed a globally high agreement rate that demonstrates our aural-visual classification’s robustness. However, some call types had low agreement rate in the RF compared to others (e.g., “gru”, “snort”, “thowp”, and “wop”). This may be due to the fact that the repertoire is composed of discrete call types and graded call types (Epp, Fournet & Davoren, 2021). Call types with low agreement in the RF may correspond to graded calls. The existence of graded calls in mother-calf groups is not surprising since a high variation is expected. These groups are composed of individuals with different attributes (young and small individual in the process of maturing its calls, adult female, and potentially an adult male) that are likely related to call characteristics (Cerchio & Dahlheim, 2001; Epp, Fournet & Davoren, 2021).

We could not establish if sounds were produced either by the mother or the calf or by any nearby conspecifics (i.e., escort) in our acoustic recordings. The accelerometer data of the tag (Goldbogen et al., 2014) could not be used to assign caller identity as the sampling rate used was 10 Hz, and even if our sampling rate was higher than 10 Hz, the close spatial proximity between a mother and her calf makes such methodology unreliable (Saddler et al., 2017). On the other hand, received levels alone are insufficient for assigning caller identity as most calls may show low amplitude levels, and several animals may be present around the tagged animal (Stimpert et al., 2020). The calls described in the present study are thus considered as the acoustic output of mother-calf groups (including possible escort). Further investigations are still needed to assign each recorded social call to an individual. We are currently planning to explore the possibility of using simultaneous deployment of Acousonde tags on the mother and the calf to determine the caller’s identity. Combining the received level of the same call on two different tags and the vertical distance between the mother and her calf (obtained from the diving profile) may allow the attribution social call to the corresponding individual.

Nine call of types out of 30 we aurally and visually identified were similar to song units recorded off the Sainte Marie island between 2013 and 2017 and were detected even in groups identified as MC, except for whine and bug sound. Assuming that mother-calf group composition did not change through the recordings’ duration, the detection of sounds similar to song units in groups composed only of a female and a calf (MC groups) suggests that female humpback whales (or even calves) are able produce sounds with similar acoustic features as males’ song units.

Some social calls recorded in mother-calf groups off Sainte Marie island presented qualitative similarities to those described in other geographic areas during the breeding, feeding seasons, or migrating routes and were assumed to be the same call type. Those calls included snort, thowp, wop (also known as whup), bark, groan, trumpet, squeak, ascending shriek, trill, and AM grunt (Dunlop et al., 2007; Dunlop, Cato & Noad, 2008; Zoidis et al., 2008; Stimpert et al., 2011; Rekdahl et al., 2013, 2017; Fournet, Szabo & Mellinger, 2015; Epp, Fournet & Davoren, 2021; Indeck et al., 2021). Given the wide range of contexts within which these calls were detected (different group types, from breeding areas to feeding areas), these social calls are probably among the most common in humpback whales, and they may have important social roles. These social calls may constitute a global repertoire shared by humpback whales around the world. Further studies are needed to determine their behavioral context and roles, especially for mother-calf pairs.

Social calls were detected even in mother-calf groups with neonate calf (C1 class), suggesting that vocal exchanges between mother and calf occur very soon right after birth. Such vocal interactions may be a way to reinforce the calf’s social bond with the mother and to imprint the calf’s voice on the mother. Calves have been reported to vocalize, and they can produce series of grunts, predominantly low-frequency sounds with a relatively narrow bandwidth (Zoidis et al., 2008). In our acoustic recordings, one call type, heek, is very similar to the amplitude modulated frequency sounds described by Zoidis et al. (2008), and thus, heek may be potentially assigned to calves.

We identified calls that can be combined or mixed with a given call type. We can assume that the calls were not successive calls from different individuals vocalizing one after the other due to the quasi absence of silence (and absence of overlap) in all instances. Call combination has not been previously described in humpback whales. Composite or concatenated calls have only been documented for few species (Koren & Geffen, 2009; Ouattara, Lemasson & Zuberbühler, 2009; Jansen, Cant & Manser, 2012, 2013; Déaux, Charrier & Clarke, 2016). Concatenated calls often have different biological functions than calls produced separately, as shown for the bark-howl vocalizations in dingo (Canis familiaris dingo) (Déaux, Charrier & Clarke, 2016).

Compared to the East-Australian catalogue (Dunlop et al., 2007) for which recordings were performed on migrating humpback whales of different group compositions (i.e., with or without calves), our repertoire contained fewer main categories (five vs. six), a similar number of call types (30 vs. 34, with eight shared call types), and a lower proportion of social calls also used in songs within the studied area (nine out of 32 in Madagascar vs. 22 out of 34 in Eastern Australia; Dunlop et al., 2007). A repertoire with 16 call types has been described for the Southeast Alaskan humpback whales, with three calls likely shared with our repertoire (Fournet, Szabo & Mellinger, 2015). The repertoire described in Newfoundland (Canada) consisted of 13 calls types and shared only one call type with our repertoire. A standardized comparative study using the same recording methods among these different areas is needed to accurately determine if a given call type is really unique to a population/area, as well as to confirm if the described call types are indeed new ones or a variation of one (previously described) call type. Our results, however, along with these previous descriptions of the repertoire of the humpback whale, support the existence of a highly diversified repertoire of social calls in a humpback whale. Our results also suggest that there is likely as much call diversity in mother-calf groups as in other social groups. Other acoustic studies focusing on mother-calf pairs also support the occurrence of such diversity (Indeck et al., 2021).

Regarding our analysis on received levels, we found that calls recorded in mother-calf groups (mother-calf pairs accompanied or not by an escort) were produced at a low amplitude level, as found previously (Tyack, 1983). The received levels ranged from 132 to 145 dB re 1μPa RMS, which is lower compared to the estimated received level of songs produced by singers of 149 to 169 dB re 1μPa (Au et al., 2006) and from our own recordings of singers (>165 dB re 1μPa, hydrophone clipped). Our results are consistent with a recent study on mother-calf pairs in Australia (136 to 141 dB re 1μPa RMS; Videsen et al., 2017). Such low-amplitude vocal production in a mother-offspring pair is quite common in mammals, such as in pinnipeds (walrus; Miller, 1966; Charrier, Aubin & Mathevon, 2010), sheep (low-pitch bleats; Sèbe et al., 2010), and cats (purring sounds; Peters, 2002). Low amplitude level in the context of mother-offspring interactions is not surprising as the communication between the mother and her calf is short- to medium-range, and the purpose is likely to maintain social contact, to reinforce the social bond and maternal attachment with the calf, and to coordinate behaviors (e.g., side by side swimming, nursing sessions, etc.). Social purpose includes specific role such as individual identification. In a short-range communication context, why yelling when talking is sufficient? Antipredator strategy and male escort avoidance may also be hypothesized to explain such low-amplitude calls (Videsen et al., 2017).

Conclusions

Our study provides a first assessment of the vocal repertoire of humpback whale mother-calf groups off Sainte Marie island, Madagascar, South Western Indian Ocean. We found that social calls recorded in these mother-calf groups are highly diversified and may be as diverse as those previously described in social groups not including calves for instance. A low acoustic intensity level characterized these social calls. The results suggest important vocal interactions between mother-calf pair, and mother-calf pairs with escorts. Our study contributes to the global catalogue of humpback whale calls and is a starting point in investigating the role of acoustic communication in humpback whale mother-calf interactions.

Supplemental Information

Supplemental Information 1 Descriptive statistics (N, mean, SD, Min and Max values) of each call type.

(*) call types qualitatively similar to previously described call types.

Click here for additional data file.

Supplemental Information 2 Raw measurements for all identified humpback whale mother-calf groups’ social calls.

Click here for additional data file.

Supplemental Information 3 Audio examples of mother-calf groups’ call types.

Click here for additional data file.

We thank everyone involved in humpback whale research in Madagascar: Cétamada team (François-Xavier Mayer, Henry Bellon, and Didier Cabocel) and all the eco-volunteers who dedicated their time and energy to this project.

Additional Information and Declarations

Competing Interests

Author Contributions

Animal Ethics

Field Study Permissions

Data Availability

The authors declare that they have no competing interests.

Anjara Saloma conceived and designed the experiments, performed the experiments, analyzed the data, prepared figures and/or tables, authored or reviewed drafts of the article, and approved the final draft.

Maevatiana N. Ratsimbazafindranahaka conceived and designed the experiments, performed the experiments, analyzed the data, prepared figures and/or tables, authored or reviewed drafts of the article, and approved the final draft.

Mathilde Martin performed the experiments, analyzed the data, authored or reviewed drafts of the article, and approved the final draft.

Aristide Andrianarimisa conceived and designed the experiments, performed the experiments, authored or reviewed drafts of the article, and approved the final draft.

Chloé Huetz conceived and designed the experiments, performed the experiments, authored or reviewed drafts of the article, and approved the final draft.

Olivier Adam conceived and designed the experiments, performed the experiments, authored or reviewed drafts of the article, and approved the final draft.

Isabelle Charrier conceived and designed the experiments, performed the experiments, prepared figures and/or tables, authored or reviewed drafts of the article, and approved the final draft.

The following information was supplied relating to ethical approvals (i.e., approving body and any reference numbers):

All methods and approaches were carried out in accordance with relevant national guidelines and regulations in force in Madagascar and were fully approved by the Ministry of Fisheries Resources, Madagascar (the approving body for all marine-related research in Madagascar) (National research and collect permits #44/13-MPRH/SG/DGPRH, #43/14-MRHP/SG/DGRHP, #28/16- MRHP/SG/DGRHP, and #26/17-MRHP/SG/DGRHP). The study complies with the European Union Directive on the Protection of Animals Used for Scientific Purposes (EU Directive 2010/63/EU).

The following information was supplied relating to field study approvals (i.e., approving body and any reference numbers):

Field studies were fully approved by the Ministry of Fisheries Resources, Madagascar.

The following information was supplied regarding data availability:

The raw measurements for all humpback whale social calls reported in the study are available in the Supplemental File.

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
