# Peer review of "Social calls in humpback whale mother-calf groups off Sainte Marie breeding ground (Madagascar, Indian Ocean)"

_PeerJ, doi:10.7717/peerj.13785_

## Round 0.1 · original submission · Major Revisions

Thank you for submitting this study to PeerJ. I regret that I am unable to accept the manuscript for publication, at least in its present form. However, I am prepared to consider a new version that carefully takes into account the suggested revisions. The reviewers liked many aspects of your study but also highlighted some important aspects for revision. These need to be addressed in detail in the new version. Such a revised manuscript may be reviewed again and there is no guarantee of acceptance. I agree that providing audio examples of important call types would enhance the manuscript. When you revise your paper, you should prepare a detailed explanation of how you have dealt with all of the reviewers' and Editor's comments.

·

Basic reporting

The article was generally well written but perhaps lacked flow - sentences were short, sharp and to the point (thank you for that!) but sometimes a little too short and sharp. I think one more read through would improve the flow. For example, try not to start two successive sentences with the same word, it tends to read like a list. Please also use quotation marks when naming sounds and refer to the spectrograms so the author can link the sound with the spectrograms.

The literature was sufficient for the field - if the field was specifically humpback whale social calls. I would be careful though when making conclusions about your work. You have a sound catalogue for humpback whales from a particular region, you can't make any inferences about how the diversity of sound types highlights the importance of acoustic interactions (last line of abstract). This comes out in the results where the sections are just lists of different sound types with some short descriptions (again referring back to short sharp sentences that ends up being a list). As a reader, it is difficult to use these results to inform my own research, first because you don't refer to the spectrograms, and second, because I am not sure what the take-home message is. See comments below in experimental design.

Spectrograms are nice, I would have liked to see the the results of the random forest presented as a figure.

The results were self-contained though I think there might have been an issue with including song sounds from nearby singers. More details is required on how you know the sounds came from the female, calf, or escort (and whether or not the escort is singing), as it is written, it suggests song sounds may have come from the female or calf (perhaps this is just a mis-communication). You suggested looking at dive profiles of tagged females and calfs to try to determine which animal made the sound - unfortunately this did not work for us. I understand you probably captured sounds from non-tagged animals, what I would suggest you make clear, is where these song sounds came from.

Experimental design

This is original research but only in the fact that it catalogues sounds produced by whales in a different region to what has been published before. The methods, and even the results, are similar/the same, to what has been done/found before. If the knowledge gap is to capture sounds made from humpback whales from that particular region at that particular time (i.e. a sound catalogue snapshot), then the knowledge gap has been filled. However, if there was a broader scope to this work, then the research question, and how this is relevant and meaningful, has not been defined.

With such a vast dataset, I recommend the authors think carefully if and how they could go beyond just another catalogue. Perhaps collaborating with others to see if, in fact, these sounds types are structurally constrained between different populations that do not meet, perhaps marrying sound production with fine-scale behaviour from the tags. The problem with the research as it is presented is that it tells us nothing new, which is a shame given the large (for humpbacks) dataset.

The investigation was mostly rigorous but I do think it needs more information on how the authors separated calls likely to be produced by the tagged animal and other close-by animals versus calls from other animals not in the group, like singers.

Validity of the findings

The rationale and benefit to the literature is not clearly stated which is why the reader is left with another description of a humpback sound catalogue. As stated above, the authors have such a good dataset, I strongly recommend they re-consider the main point of this study or at least point out why it is of benefit to the broader scientific community that this study be published, and read.

The conclusions do not link to the original research question as it needs to be more clearly defined. In addition, the results presented here do not support the conclusions made - you say that female-calf repertoires may be as diverse as other groups but no comparison was made of repertoire size between different group compositions, with the received levels you report, are you sure these calls came from the tagged animal, and even then the received levels haven't been compared with other cohorts so you can't say they were "low" - to do this you would need to compare like with like - same tags, same environment, same method. The results don't support the conclusion that they should important vocal interactions between female-calfs and other adults as you haven't demonstrated this behaviourally, they do, however, add to the global catalogue, but again it would be good to see a reasoning for this.

·

Basic reporting

This study examined the social calls of humpback whales in cow-calf dyads in Madagascar. I was extremely impressed by this study, and it was a pleasure to read through it all. It will be a great contribution to the bioacoustics literature, and I only have minor suggestions to improve this work. The introduction clearly set the scene by introducing previous literature and identifying the knowledge gaps. The methods and results were well documented with only minor amendments needed (see below). The discussion was comprehensive and clearly written. The English language should be edited to improve readability. I have made comments below to assist you with this, but a general copy edit is also advised. Some minor clarification is also needed for the methods.

Introduction:
Line 47 – Replace beginning of sentence with “The term social sounds is a general expression…”
Line 58 – What is the difference between lone adults and singletons? Please clarify or remove one of these terms.
Line 73 – single quotations around ‘principal escort’
Line 83 – Replace phrase with “….. and thus further investigations are needed to accurately assign…”
Line 90 – replace second ‘complement’ with “to contribute to the global…”
Line 90 – correct spelling is catalogue

Experimental design

Materials and methods:
Please provide the animal ethics statement in the first paragraph of the materials and methods text.

All figures – you have mentioned that you generated the spectrograms using Seewave but this is not described in the methods so please add it here too. Also cite Seewave.

Line 100 – remove dash after 6.40 and 5 m.
Line 103 – please clarify whether the staff team and role of each member remained the same for all of the taggings
Line 133 – Correct English is “Calls that were clearly audible…”
Line 138 – Replace with “New names were assigned for the remaining call types onomatopoeically, as the behavioural context or biological function of these new call types is still unknown”
Line 144 – Mention the specific number of temporal and spectral acoustic parameters analysed
Line 144 to line 163 – to improve readability, some of this information could be provided in table format and referred to within this paragraph of text.
Line 164 to line 177 – this should go under a separate subheading called statistical analysis
Line 178 – Does received level refer to the sound intensity? I would recommend moving it under the acoustic analysis subheading since it relates to that and then move the statistical analysis section after it.

Validity of the findings

Results:
Line 192 – Three groups had a C1 calf, three had a C2 calf and 15 had a C3 calf.
Line 200 – Sentence starting with eleven – reference to previously described call types is more of a discussion point so isn’t appropriate for the results section. Can you instead give a brief overview of how they are characterised?
Line 236 – creak not creek
Line 236 and line 237 – Sentences on groan, wiper and creak - These sentences can be merged to improve readability
Line 262 – “Door was heard in an MC group”

Discussion:
Line 291 – Indeed is not necessary in this sentence.
Line 300 – simultaneous deployment of what?
Line 304 to 306 – recordings from the other days off - is this a personal communication? Have you documented these findings anywhere? Is it from a previous study?
Line 340 – five versus six. As a general rule, if number < 10, write out in words
Line 345 – Replace with “The repertoire described in Newfoundland Canada…”
Line 358 – 149 to 169 dB
Line 364 – what about communication for identification purposes?
Line 374 – Replace with “between mother calf pairs, and mother calf pairs with escorts…”
Line 374 – Catalogue not catalog
Line 375 – Replace with “and is a starting point” – can be more assertive.

Figures:
Some English corrections needed, see below:
Figure 1 – “Most of the LF sounds…… except for booms and snorts….”
Figure 2 – “Downsweep, woohoo and trump calls…”
Figure 2 – “Heeks were produced in association with Grus (LF)…”
Figure 3 – “Squeaks were very short calls…. ascending shrieks were one of the longest….”
Figure 4 – “Spectrograms of amplitude-modulated sounds..”
Figure 4 - “Except for AM grunts, AM sounds are long calls ranging from 1 to 5 seconds…” “Whine, trill and bug sounds were also…”
Figure 5 - “Spectrogram of pulsed sounds (PS).

Tables:
Table 1 – please provide the units for the analyzed recording duration.
Table 4 - remove the first ‘and’ from the sentence
Table 4 – Replace calls with call category or call type to align with what is mentioned in the body of text
Table 4 – amplitude received level – is this the mean +/- SD? If so, please specify in the table caption or as a superscript
Table S1 – next to the snort, thwop, wop, bark, groan, trumpet, creak, squeak, ascending shriek, trill, and AM grunt variables there is a star, what does this refer to? There is no superscript describing what it is for.
Table S1 – ‘thwop’ not ‘thowp’ and ‘creak’ not ‘creek’

Reviewer 3 ·

Basic reporting

This is an interesting contribution; studying whales is challenging, and therefor I belief that each contribution is highly valuable.
I think the authors can provide more background on humpback whale vocal behavior in general. For readers unfamiliar with that topic, it is particularly in the beginning hard to follow; I was a bit confused by the term call types, call categories, ect. Later on it became clear, but the introduction needs more background. Also, in my opinion, you should highlight and discuss more the fact that humpback whales are vocal learners. This is special, and it might not be limited to male songs; calve might have to learn their calls, or aspects of them, and I really think that this should be addressed; in the introduction, but maybe also in the discussion.
Also, it would be good to briefly state if there is knowledge about mother calf vocal behaviour of other baleen whales. and if yes, report it.

Be more cautious in the general discussion, particularly with call type definition; if you cannot identity the caller (which is clearly difficult in humpback whales), it is hardly possible to define call types; calf calls might differ from adult calls due to maturational processes, but it might still be the same call type; in case of the MCE groups; if you cannot define the caller, you should be very careful with defining new call types (sex differences?).

I really appreciate the spectrograms, this is so important for an acoustic paper. but it would be further fantastic if you could also provide sound examples to listen to as supplemental material.

Experimental design

The methodology is very interesting and I appreciate that you state the advantages but also the limits of the equipment/method. It is well described and the equipment used is of high technical standard. Also, to the best of my knowledge, it seems that the ethical standards have been met.

The aims were clearly outlined, and the method used was appropriate; This includes also data analyses; (acoustic data analysis as well as statical analyses is sound).

Validity of the findings

The authors provide tables, and the most important underlying data as supplementals. As mentioned before, it would be good if the authors could provide audio-files as supplemental material, so that the reader can also listens to some sounds. This will not just be highly interesting, it will also be an important contribution for further comparative studies (which are important as the authors state in the manuscript).

As mentioned before, we need more information of how established the vocal repertoire for humpback whales is in general; are different authors agree on the call types, and the terminology? the authors mention in the discussion that differences in the reported vocal repertoire might be due to differences in methodology; does this address how recordings were made, or also the analysis; acoustic as well as statistical analysis used? be more specific and give more information.
I know I am stressing this, but we need more background information.

Additional comments

more specific comments with line numbers:
Abstract line 24: why do you write “socials calls are likely produced by all types …”
In the introduction you cite literature, so maybe consider rephrasing because it reads as if it is not known whether social calls are produced or not. And in line 58 you explicitly state that individuals in all group compositions produce social calls.

Line 57: is it possible to explain the sound generator in a bit more detail?

Line 128: can you provide more information of how call types are classified; what parameter were used; and based on what statistical method? Or is that mainly based on human visual and auditive perception. How discrete or graded are those call types; what about within call type variability? I would suggest to give more information on that, and if it is not known, also explicitly state this;
We need to understand how valid the knowledge about the vocal repertoire or the number of call types is. Because I am a bit confused; so are social call types categorized as LF, MF, or HF, or as call types per se;

Line 202: what does it need for a call type to be eligible?

Line 307: just a thought; but might it be that male calves do produce these “song-like” calls?

Line 321: could you suggest what would be the main purpose of mother calf communication? Keeping contact, coordinating suckling, ect.

Line 331: how do you know that one individual concatenates the calls; can you be sure that there are not calls from two individuals?

---

## Round 0.2 · Major Revisions

Before a decision can be made on this study, we require further information and clarity regarding the ethics of the research.

See the guidance on Animal Research provided by PEERJ. https://peerj.com/about/policies-and-procedures/#animal-research Please follow the PEERJ guidance.

Please try to avoid vague statements such as, "All methods were carried out following relevant local guidelines and regulations. Editors, reviewers and readers cannot be expected to search for the relevant regulations in France and/or Madagascar.

Has the research been reviewed and approved by a university animal ethics committee?

Please group information on ethics together in the Methods, so text in lines 186-188 should be earlier.

---

## Round 0.3 · accepted · Accept

Really interesting initial study of humpback whales in the Indian Ocean.